# Mullite-Fibers-Reinforced Bagasse Cellulose Aerogels with Excellent Mechanical, Flame Retardant, and Thermal Insulation Properties

**DOI:** 10.3390/ma17153737

**Published:** 2024-07-28

**Authors:** Shuang Wang, Miao Sun, Junyi Lv, Jianming Gu, Qing Xu, Yage Li, Xin Zhang, Hongjuan Duan, Shaoping Li

**Affiliations:** 1The State Key Laboratory of Refractories and Metallurgy, Wuhan University of Science and Technology, Wuhan 430081, China; shuangwang@wust.edu.cn (S.W.); miao740526@163.com (M.S.); lvjunyi@wust.edu.cn (J.L.); gjm15503921886@163.com (J.G.); xuqing99@yahoo.com (Q.X.); liyg@zjweu.edu.cn (Y.L.); 17634809427@163.com (X.Z.); 2Hubei Three Gorges Laboratory, Yichang 443007, China; lisp@xingfagroup.com

**Keywords:** cellulose aerogels, mullite fibers, thermal insulation, mechanical properties, flame retardant

## Abstract

Cellulose aerogels are considered as ideal thermal insulation materials owing to their excellent properties such as a low density, high porosity, and low thermal conductivity. However, they still suffer from poor mechanical properties and low flame retardancy. In this study, mullite-fibers-reinforced bagasse cellulose (Mubce) aerogels are designed using bagasse cellulose as the raw material, mullite fibers as the reinforcing agent, glutaraldehyde as the cross-linking agent, and chitosan as the additive. The resulted Mubce aerogels exhibit a low density of 0.085 g/cm^3^, a high porosity of 93.2%, a low thermal conductivity of 0.0276 W/(m∙K), superior mechanical performances, and an enhanced flame retardancy. The present work offers a novel and straightforward strategy for creating high-performance aerogels, aiming to broaden the application of cellulose aerogels in thermal insulation.

## 1. Introduction

The global energy crisis is becoming increasingly severe. Specifically, the amount of coal and oil extracted increased from 6 billion tons to 15 billion tons, nearly doubling in just six years [1]. The construction industry is one of the most critical sectors for energy conservation and emission reduction, which accounts for about 35% of energy consumption [2]. Therefore, it is of practical significance to develop an energy-saving and environmentally friendly insulation material [3,4]. Aerogel materials have excellent thermal insulation properties, making them a research hotspot in the field of thermal insulation materials. The advantages of a low density, high porosity (>90%), and low thermal conductivity mean that aerogel can be widely used in energy storage, thermal insulation, biomedicine, and many other aspects [5,6,7]. Cellulose emerges as a promising alternative for preparing aerogels given its abundant, renewable, and sustainable nature. Nevertheless, its broader application in thermal insulation is curtailed due to inadequate flame retardancy [8,9] and poor mechanical properties [10]. Notably, cellulose materials can absorb water molecules and form hydrogen bonds at high relative humidity environments, leading to the relaxation of their molecular structure and a reduction in their mechanical properties [11]. This, therefore, presents a challenge and an opportunity to design a straightforward method for producing cellulose aerogels with superior elasticity, fatigue resistance, and enhanced flame retardancy.

Enhancing the elasticity of cellulose aerogels is feasible through the construction of cross-linked network structures by various chemical processes, including amidation [12], a Schiff base reaction [13], a thiol–alkene reaction [14], and ring-opening reactions [15], alongside the introduction of inorganic fibers. For instance, Zhang et al. [16] synthesized highly elastic cellulose aerogels by crosslinking cellulose and chitosan with glutaraldehyde, which maintained structural integrity even under 60% strain and exhibited a compressive strength of 194.40 kPa at 50% strain. Mi et al. [17] enhanced the modulus of elasticity and compressive properties of cellulose aerogels by incorporating silica fibers, enabling the aerogels to fully recover from 50% compressive strain. Similarly, Yi et al. [18] fabricated a cellulose aerogel with an excellent fatigue resistance by designing the multiscale structure of aluminosilicate fibers/montmorillonite whiskers to strengthen the interfacial bonding, and the aerogels could reach 97.8% of the initial stress even after 200 cycles of deformation with large strain.

In order to overcome the problems associated with the poor flame retardancy of aerogels, the introduction of inorganic flame retardants has received widespread attention and has proved to be an effective strategy in many studies [19]. For example, Luo et al. [20] prepared aerogels with an excellent flame retardancy using Mg/Al-layered double hydroxide as a flame retardant; the peak heat release rate (pHRR) of the resultant composite aerogel was reduced by 41% and the smoke production rate was decreased by 79% compared with pure cellulose aerogel. However, the fabrication process of the aerogel was relatively complex. SiO_2_ nanoparticles are spherical nanomaterials of an extremely small scale. Owing to the hydrophobicity of their molecules, silica nanoparticles are often used as additives into aerogels to control their shrinkage and pore size, thus enhancing the flame retardant and mechanical properties [21,22]. Yang et al. [23] successfully developed an aerogel by an effective coating of SiO_2_ nanoparticles on sunflower stalk fibers. The strategy successfully hindered combustible molecules from burning and retarded heat diffusion from interfacial to the interior, and thus the residual carbon content in the modified aerogels was increased by 31% compared to unmodified samples. However, the flame retardancy failed to be dispersed homogeneously in the cellulose suspension, leading to a relatively dense structure in the prepared aerogel.

To meet the above requirements, mullite fibers (MF) were used to reinforce bagasse cellulose (BCE), creating mullite-fibers-reinforced bagasse cellulose (Mubce) aerogels with exceptional mechanical properties, flame retardancy, and thermal insulation capabilities. Bagasse was used as the raw material and glutaraldehyde was used as a cross-linking agent to form strong covalent bonds with cellulose and chitosan to avoid the structure collapse of the samples during the atmospheric pressure drying process. As a result, the Mubce aerogels prepared demonstrated extremely low thermal conductivities, robust elasticity, and fatigue resistance. Additionally, the flame-retardant properties of the composite aerogels were enhanced compared to BCE aerogels. Therefore, this work is expected to offer a novel and inspiring idea for the large-scale production and wide application of cellulose aerogels and the alleviation of global energy shortages.

## 2. Materials and Methods

### 2.1. Raw Materials

Bagasse (cellulose: 36 wt %, hemicellulose: 38 wt %, and lignin: 4.9 wt %) was obtained from Guangxi Yuante Agricultural Technology Co., Ltd. (Nanning, China), and MF with a fiber diameter of 7–11 µm were obtained from Zibo Bin Hong Insulation Materials Co., Ltd. (Zibo, China), as illustrated in Figure 1a. The chemical reagents, including aqueous solution of glutaraldehyde (purity of 25 wt%), sodium hydroxide powder (NaOH, AR), 80% sodium chlorite (NaClO_2_, purity of 80 wt%) powder, and analytical grade acetic acid (purity of 99.5 wt%), were all purchased from Shanghai Macklin Biochemical Technology Co., Ltd. (Shanghai, China). Chitosan, with a viscosity between 200–400 MPa·s and a deacetylation degree of ≥95%, was supplied by Shanghai Aladdin Reagent Co., Ltd. (Shanghai, China). Acetic acid played a crucial role in facilitating the dissolution of chitosan and enhancing the viscosity of the mixture.

### 2.2. Preparation of BCE Aerogels and Mubce Aerogels

Based on our previous research [16], cellulose was extracted from bagasse. Subsequently, a 60 g chitosan solution (2 wt% concentration, corresponding to 40 wt% of the BCE mass) was combined with a 100 g cellulose suspension (3 wt% cellulose content), under mechanical stirring at 500 rpm.

After achieving a homogeneous mixture of the chitosan solution and cellulose suspension, 1.6 g of glutaraldehyde solution (representing 1 wt% of the total system mass) was added. The mixture was vigorously stirred for 30 s before being promptly transferred into a mold and frozen at −18 °C for 48 h. The resultant wet gel was then air-dried at room temperature over another 48 h to yield the BCE aerogels.

Figure 1b delineates the fabrication process of Mubce aerogel: Initially, varying amounts of MF (50 wt %, 100 wt %, 150 wt %, and 200 wt % relative to BCE mass) and 40 g of chitosan solution (chitosan content of 40 wt % relative to BCE mass) were added to a 100 g BCE suspension (with 2 wt % cellulose content). Following a thorough mixing of the chitosan solution and cellulose suspension, 1.4 g of glutaraldehyde solution (1 wt % relative to the total system mass) was added.

After stirring briskly for 30 s, the slurry was immediately poured into a mold and frozen at −18 °C for 48 h. The wet gels were subsequently air-dried at ambient temperature for 48 h, resulting in the formation of Mubce aerogels, which were designated as Mubce-50, Mubce-100, Mubce-150, and Mubce-200, respectively, based on the proportion of MF included.

### 2.3. Characterization

The volume density (ρ) of the prepared aerogel was calculated using formula (1).
(1)$$\rho =\frac{m}{v}$$
where *m* is the mass of the aerogel and *v* is the volume of the aerogel.

The compression test for the aerogel samples was performed in accordance with the ISO 844:2007(E) standard using a universal testing machine (Model TSE502A, Wance Testing Machine Co., Ltd., Wuhan, China) at a compression speed of 20 mm/min. The dimensions of the test samples were roughly 30 mm in diameter and 20 mm in height. Pore size distribution and porosity were obtained by a mercury intrusion porosimetry (AUTOPORE 9500, McMurritik Instruments, Shanghai, China). Infrared thermal imaging of the samples was carried out with an infrared thermal imager (Fluke TiX660, Fluke, Everett, Washington, DC, USA) to record surface temperatures. The microstructural analysis was performed using a field emission scanning electron microscope (SEM, Nova Nano 400, c, Hillsboro, OR, USA). The acceleration voltage was set to 10 kV and gold sputtering was used. Elemental analysis was conducted with an X-ray photoelectron spectrometer (XPS, AXIS SUPRA+, Shimadzu Ltd., Kyoto, Japan). The chemical structure was investigated using a Fourier transform infrared (FTIR) spectrophotometer (INVENIO-R, Bruker, Bremen, Germany). Thermogravimetric analysis (TGA) was carried out on a thermal analyzer (STA 449C, NETZSCH, Selb, Germany) with a heating rate of 10 °C/min in an argon atmosphere from 30 to 800 °C. The thermal conductivity of the aerogels was measured in −40 to 150 °C, utilizing an infrared thermal conductivity tester (TPS 2500S, Hot Disk, Sweden) in line with the ISO 22007-2 standard. The Limiting Oxygen Index (LOI) was assessed using an oxygen index tester (FTT0077, FTT, London, UK) following the ISO 4589 standard, with sample dimensions of 80 × 10 × 4 mm^3^. Combustion characteristics were evaluated via a cone calorimeter (FTT00077, FTT, UK) according to ISO 5660 standard, with testing sample size of 100 × 100 × 4 mm^3^ under a radiant heat flux of 35 kW/m^2^. All raw data were processed using origin software.

## 3. Results

### 3.1. Preparation and Characterization of Mubce Aerogels

The ambient pressure drying method is adopted to fabricate high-performance Mubce aerogels. Figure 1c illustrates that some MF are encapsulated within the BCE, as marked by the yellow arrow, while others are nestled between layers of BCE, as pointed out by the blue arrow. Compared to the untreated MF, the aspect ratio of the MF within the prepared aerogels was relatively low (3–32), which was ascribed to aggregation and or cutting short, arisen from the shear forces during magnetic stirring. The introduction of MF in the BCE will enhance the elasticity and fatigue resistance of the final aerogel. Using various molds, aerogels of different shapes in a wood-yellow hue can be easily crafted (Figure 1d). Additionally, as shown in Figure 1e, these aerogels can be delicately placed on flower petals without compromising the structure of the petals. Figure 1f presents the pore size distribution of the aerogels measured by mercury intrusion porosimetry, revealing an average pore size of 58.9 μm.

The FTIR spectra, depicted in Figure 2a, shows distinct peak characteristics of both BCE and the Mubce-150 aerogels. Notably, key peaks corresponding to Mubce-150 aerogels, such as the -OH stretching vibrations in the range of 3200–3500 cm^−1^, C-H asymmetric stretching vibrations around 2950–3000 cm^−1^, C=C stretching vibration peak at 1600 cm^−1^, and C-O stretching peaks at 1030 cm^−1^ were observed in the spectra. Additionally, the emergence of new peaks at 563 cm^−1^ in the composite aerogels was likely attributable to the vibrations of six-coordinated Al-O bonds, indicating the successful introduction of MF [24,25,26,27,28]. Compared to the BCE aerogels, the peaks of the Mubce aerogels showed a slight blue shift. This shift may be due to the formation of hydrogen bonds between the bound water on the MF surface and the -OH groups in the BCE aerogels, which decreased the wavenumber of the -OH absorption band of the Mubce aerogels.

The XPS spectrum of the Mubce aerogels, as shown in Appendix A, reveals peaks for C1s, O1s, N1s, Si 2p, and Al 2p. The presence of the Si 2p and Al 2p peaks confirms the successful incorporation of MF, corroborating the FTIR results. The C1s spectra of BCE aerogels, presented in Figure 2b, can be deconvoluted into five distinct components at binding energies of 284.44 eV, 285.14 eV, 286.35 eV, 287.68 eV, and 288.15 eV, corresponding to C-C, C-N, C-O, C=N, and C=O, respectively. The C1s spectrum of Mubce aerogels, as shown in Figure 2c, displays no new peaks compared to BCE aerogels, confirming the existence of similar species in both aerogels. The N1s spectra of BCE and Mubce aerogels, depicted in Figure 2d,e, each show four component fits corresponding to N-H (402.3 eV), N-C (400.9 eV), -NH_2_ (399.6 eV), and C-N=C (398.5 eV), with the first three derived from chitosan and the last attributable to a reaction between chitosan and glutaraldehyde.

The XRD of MF, BCE aerogels, and Mubce-150 aerogels was carried out. The XRD pattern of the Mubce-150 aerogels demonstrated distinct peaks corresponding to MF (JCPDS No. 00-038-0471), confirming the existence of MF within the BCE (in Figure 2f). Moreover, the EDS element mappings of the Mubce-150 aerogels, as depicted in Appendix A, showed the presence of elemental Al and Si originating from MF, further confirming the successful incorporation of MF into the Mubce aerogels.

SEM of BCE and Mubce-150 aerogels was performed to investigate their microstructures. The BCE aerogels exhibited a relatively dense three-dimensional porous structure, as shown in Figure 2g, which can be attributed to the entanglement of cellulose constraining the growth of ice crystals [29]. After adding MF to the BCE suspension, the as-prepared Mubce aerogels formed a lamellar structure (in Figure 1c), where some MF were encapsulated in the BCE while others were nested between the BCE layers. The even dispersion of the MF within the BCE resulted from the good compatibility between MF and the BCE suspension, and the disordered, layered structure was attributed to the random growth of ice crystals in the refrigerator (−18 °C). As shown in Figure 2g–k, the layer space of the aerogel expanded with the increase in the MF concentration, suggesting that the MF encapsulated within the BCE effectively reduced the stress concentration during the drying process. This will enhance the elasticity and fatigue resistance of the as-prepared Mubce aerogels. Additionally, the MF nested between the BCE layers served as pillars, not only expanding the layer space, but also leading to an increase in the compressive strength of the aerogels (in Figure 2l).

### 3.2. Mechanical Properties of Mubce Aerogels

Mubce aerogel can be used in a wide range of applications, owing to its improved compressive strength, elasticity, and fatigue resistance. As shown in Figure 3a, the stress–strain curves for both BCE and Mubce aerogels exhibited similar shapes.

Furthermore, the curves showed no obvious plastic yield plateau at strains less than 60% (ε < 60%), and transformed into a densification stage with a sharp increase in stress at ε > 60%. The compressive strength and density of the as-prepared Mubce aerogels increased significantly with the introduction of MF compared to BCE aerogels in Figure 3b. Specifically, the compressive strength of the Mubce-150 aerogels increased to 272 kPa (at 60% strain), which was 1.3 times that of the BCE aerogels (compressive strength of 208 kPa). Moreover, the Mubce-150 aerogels were able to withstand 1561 times their own weight without significant deformation, verifying the effective reinforcing effect of MF, as showed in Figure 3c. However, further increasing the MF content to 200 wt %, where the compressive strength of Mubce aerogels decreased again, may be attributed to the aggregation of MF at a higher concentration. It is believed that the aggregation resulted in an increase in the stress concentration within the aerogels during the drying process [30].

Moreover, Figure 3d presented the stress–strain curves of Mubce-150 aerogels at different compressive strains of 20%, 40%, and 60%. The aerogels fully recovered as the strain increased from 20% to 60%, showcasing their remarkable elasticity. To further assess the fatigue resistance of the aerogels, a cyclic compression test at 40% strain was conducted. Figure 3e,f indicated that even after 150 cycles, the Mubce-150 aerogels retain 92.0% of the initial stress, an 11.27% improvement over BCE aerogels. Additionally, the stress in Mubce-150 aerogels did not significantly decrease even after 500 cycles, highlighting the crucial role of MF in enhancing the elasticity and fatigue resistance of the aerogels. The mechanical properties of these aerogels are superior or comparable to those of most previously reported biomass-derived cellulose aerogels, as listed in Appendix A. Even under extreme conditions such as liquid nitrogen temperatures (−196 °C), Mubce-150 aerogels maintain their elasticity, as demonstrated in Appendix A, highlighting the widely applicatied temperature range of the as-prepared aerogels. Overall, the excellent elasticity and fatigue resistance of the Mubce aerogels benefit the long-term stability of the cellulose aerogels in practical applications.

### 3.3. Thermal Insulation and Fire Resistance Properties of Mubce Aerogels

As illustrated in Figure 4a, the Mubce-150 aerogels showcased an outstanding thermal insulation performance, with a thermal conductivity ranging from 0.0257 to 0.0297 W/(m∙K) across a temperature span of −40 to 150 °C. Owing to its low thermal conductivity, the addition of the MF can reduce the overall thermal conductivity of the aerogels. Additionally, the pore structure of the aerogels was changed by MF, making the pores smaller and more complex. This effectively reduced the thermal conduction by the gas phase. Furthermore, the MF has excellent infrared reflection capabilities, further decreasing heat transfer via radiation. As depicted in Figure 4b, the thermal conductivity of the Mubce-150 aerogel is significantly lower than that of most other aerogels in the previously published literature [24,31,32,33,34,35], affirming its superior insulation capabilities. This strategy will also inspire further research into the development of fiber-reinforced functional aerogels and their applications in thermal insulation.

To further evaluate the insulation performance of Mubce aerogels, the as-prepared Mubce-150 aerogels were placed on a heating stage with a temperature of 160 °C for 60 min, and their upper surface temperature was monitored every 5 min using an infrared thermal imager. As illustrated in Figure 4c, the upper surface temperature of the aerogel gradually increased to 34.3 °C before stabilizing at 33.8 °C. Notably, the temperature difference between the upper and lower surfaces of the aerogels reached an impressive 125.8 °C, demonstrating a superior insulation performance compared to most aerogels reported in the literature, as detailed in Appendix A. The strategy for the preparation of Mubce aerogels not only offers a highly effective approach to convert bagasse into high-added-value products, but also introduces a novel concept for the widespread utilization of cellulose aerogels.

Thermogravimetric analysis (TGA) under an argon atmosphere was employed to evaluate the impact of MF on the thermal stability of the aerogels. Appendix A showed that the initial weight loss observed in the TGA curve, ranging from 30 to 100 °C, was attributed to the removal of adsorbed and bound water from the sample [36]. As shown in Appendix A, compared to BCE aerogels, the temperature at which the Mubce-150 aerogels experienced a 10% mass loss (T_−10%_) increased from 235.5 °C to 255.4 °C. Additionally, the temperature for a 50% mass loss (T_−50%_) rose to 458.4 °C, exhibiting a 31.2% increase. The maximum thermal decomposition rate of the Mubce-150 aerogels was lower than that of the BCE aerogels (Appendix A), indicating that the thermal stability of the cellulose aerogel was improved by the addition of MF. This may be because MF formed a physical barrier within the aerogel, which can slow down the heat transfer and the release of decomposition products. Moreover, due to the high thermal stability and mechanical strength of MF, their addition can enhance the overall mechanical performance of the aerogel, making the structure hard to collapse at high temperatures. At 800 °C, the residue of Mubce aerogels was 42.3%, which was 69.2% higher than that of BCE aerogels. Furthermore, the Limiting Oxygen Index (LOI) test was conducted to verify the exceptional flame retardancy of Mubce-150 aerogels. The LOI of the composite aerogel was 30.6%, which was 1.3 times that of the previously reported cellulose aerogels [37]. Materials with an LOI over 26% are generally considered hard to ignite [38]. In a word, the addition of MF significantly enhances the flame retardancy of cellulose aerogels.

Figure 4d,e and Appendix A visually demonstrate the protective effect of Mubce-150 and BCE aerogels on paper flowers under butane gas combustion. The BCE aerogel only protected the paper flowers for 19 s, whereas the Mubce-150 aerogel extended the protection to over 45 s, verifying that the flame retardancy of the aerogel was effectively improved by MF. Compared with areas distant from the flame (Figure 4g), the carbonized areas in the Mubce aerogels (Figure 4h) had no significant changes and maintained the structural integrity. In the ablation zone (Figure 4i), although most of the cellulose underwent thermal decomposition and oxidation, the MF had not significantly changed, effectively slowing the spread of the flame.

The cone calorimetry was also conducted to further evaluate the flame retardancy of BCE and Mubce aerogels under real fire conditions. As illustrated in Figure 4j, the pHRR of the Mubce-150 aerogels was 100.45 kW/m^2^, which was 42.08% lower than that of the BCE aerogels (173.44 kW/m^2^). During the initial 80 s, the THR for the BCE aerogels reached 3.96 MJ/m^2^, while the Mubce aerogels reduced to 3.49 MJ/m^2^, as shown in Figure 4k. MLR refers to the mass loss rate of the sample during combustion and serves as a critical parameter for evaluating the combustion behavior of materials. The MLR of the Mubce-150 aerogels was obviously lower than that of the BCE aerogels, demonstrating again the superior flame-retardant property of the former (Figure 4l). Moreover, the time taken for the composite aerogel to lose 55% of its weight was 78 s, which was about five times higher than that of the BCE aerogel (Appendix A), demonstrating again that the incorporation of MF markedly boosts the flame-retardant performance of cellulose aerogels.

## 4. Conclusions

In summary, Mubce aerogels with exceptional mechanical strength, flame retardancy, and thermal insulation were successfully developed using an ambient temperature and air-drying method. MF played a crucial role in decreasing cellulose entanglement, enlarging the layer space within the aerogels, and reducing stress concentrations. The resultant Mubce aerogels exhibited a compressive strength of 272 kPa (at 60% strain) and exceptional fatigue resistance. Additionally, the aerogels demonstrated a high porosity of 93.2% and maintained a remarkably low thermal conductivity ranging from 0.0257 to 0.0297 W/(m∙K) across a temperature range of −40 to 150 °C. Even when heated at 160 °C for 60 min, the temperature difference between the upper and lower surfaces of the as-prepared Mubce aerogels was as high as 125.8 °C, affirming their outstanding thermal insulation performances. With an LOI of 30.6%, the aerogels further displayed enhanced flame-retardant properties. Hence, the innovative approach employed in the fabrication of Mubce aerogels not only offers fresh inspiration for the high-added-value utilization of agricultural waste, but also facilitates the widespread application of cellulose aerogels under extreme conditions. Additionally, potential future research directions should focus on improving the drying technology to further shorten the production cycle and optimizing the extraction techniques of cellulose from agricultural waste to effectively utilize lignin and hemicellulose.

## Figures and Tables

**Figure 1 materials-17-03737-f001:**
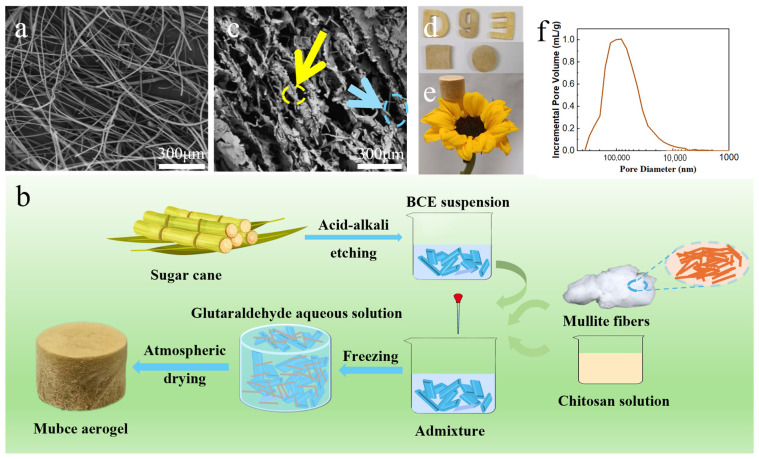
(**a**) SEM image of raw MF. (**b**) Schematic illustration of the fabrication process of Mubce aerogels. (**c**) SEM image of as-prepared Mubce-150 aerogels (MF are encapsulated within the BCE, as marked by the yellow arrow, while others are nestled between layers of BCE, as pointed out by the blue arrow). The optical image of (**d**) Mubce-150 aerogels with different shapes and (**e**) the photo of the aerogel standing on a petal. And (**f**) pore size distribution curve of Mubce aerogels obtained by mercury intrusion porosimetry.

**Figure 2 materials-17-03737-f002:**
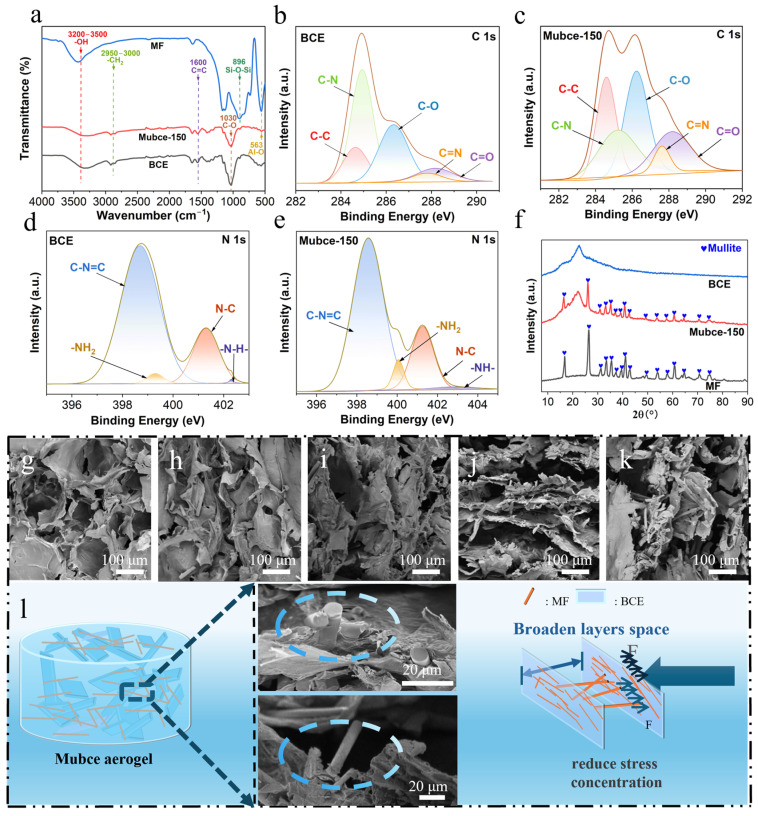
(**a**) FTIR spectra of MF, BCE aerogels, and Mubce-150 aerogels; C 1s survey spectra of (**b**) BCE and (**c**) Mubce-150 aerogels. N 1s survey spectra of (**d**) BCE and (**e**) Mubce-150 aerogels; (**f**) XRD pattern of BCE and Mubce-150 aerogels; SEM images of (**g**) BCE aerogels, (**h**) Mubce-50 aerogels, (**i**) Mubce-100 aerogels, (**j**) Mubce-150 aerogels, and (**k**) Mubce-200 aerogels. And (**l**) schematic diagram of enhanced mechanical properties of BCE aerogels.

**Figure 3 materials-17-03737-f003:**
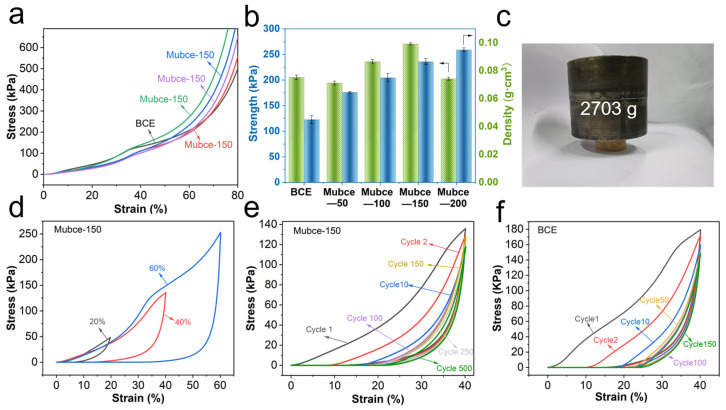
(**a**) Stress–strain curves of BCE and Mubce aerogels. (**b**) The corresponding compressive strength and density. (**c**) Photographs of Mubce-150 aerogel withstanding 1561 times its own weight. (**d**) Stress–strain curves of Mubce-150 aerogels at different compressive strains. Fatigue test of (**e**) Mubce-150 aerogels and (**f**) BCE aerogels under 40% compressive strain.

**Figure 4 materials-17-03737-f004:**
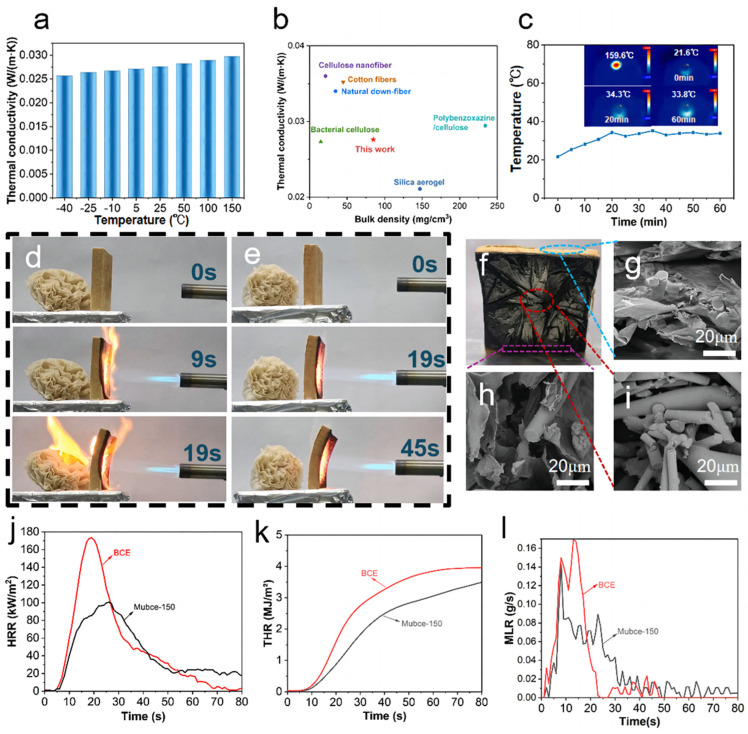
(**a**) Thermal conductivity of Mubce-150 aerogels at different temperatures. (**b**) Comparison of the volume density and thermal conductivity of Mubce aerogels with other reported aerogels [24,31,32,33,34,35]. (**c**) Temperature variation curve of the upper surface of the samples heated at 160 °C for 60 min and infrared images of the samples at different heating times on a heating stage. Optical photographs of (**d**) BCE and (**e**) Mubce-150 aerogels heated by butane torch flame (aerogel specimen sizes are 90 × 90 × 15 mm^3^). (**f**) Optical image of Mubce-150 aerogels under butane flame for 45s. SEM iages of Mubce-150 aerogels: (**g**) area far away from the butane torch flame, (**h**) carbonized region not directly in contact with the butane torch flame, and (**i**) ablation region in direct contact with the butane torch flame. And (**j**) Heat Release Rate (HRR), (**k**) Total Heat Release (THR), and (**l**) Mass Loss Rate (MLR) curves of BCE and Mubce-150 aerogels.

## Data Availability

Data are contained within this article.

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
