# Peer review of "Mullite-Fibers-Reinforced Bagasse Cellulose Aerogels with Excellent Mechanical, Flame Retardant, and Thermal Insulation Properties"

_materials, 2024, doi:10.3390/ma17153737_

Round 1

Reviewer 1 Report

Comments and Suggestions for Authors

Aerogels are increasingly becoming the subject of scientific research. To obtain them, both synthetic and natural polymers are used. In this work, it is proposed to use bagasse cellulose and mullite fibers to obtain aerogels.

The introduction in the manuscript is, in my opinion, short, based on 15 references.

63-69. When formulating the goal, the authors provide the obtained values, which is not required. If authors want to put forward a hypothesis, then it is not necessary to provide values. I recommend clearly formulating the purpose of the work and preferably putting forward a hypothesis.

When describing materials, it is advisable to add their characteristics. And when describing methods, it is better to divide them into separate paragraphs.

144. I suggest rounding the pore size to tenths or it is necessary to indicate a statistical error.

The quality of the spectra and diffraction patterns in Figure 2 needs to be improved.

There is no need for Figure S3; Figure 2 already contains a micrograph of the BCE airgel.

Where did the authors get the data for Figure 4b? If from literary sources, then they must be indicated.

Why does the temperature at which the maximum rate is observed coincide for two samples with different rates of mass loss (Figure S4)? It is also not clear why the TGA experiment was carried out in an inert environment, and not in air or oxygen?

266. Due to what does thermal stability increase?

Conclusions briefly and clearly reflect the results obtained.

The manuscript is worth reading, but I feel it lacks sufficient discussion of the findings. For example, it is not clear why, with less porosity, the thermal conductivity is at the level of samples with more porosity.

Author Response

According to your valuable suggestions, we have revised the manuscript. Please see the attached file for details.

Reviewer 2 Report

Comments and Suggestions for Authors

The manuscript titled “Mullite fibers reinforced bagasse cellulose aerogels with excellent mechanical, flame retardant and thermal insulation properties” is a work where the authors interrogated the morphology, mechanical and thermal performance of aerogels made of reinforce bagasse cellulose and reinforced with mullite fibers. The most relevant outcomes found in this work can be interesting for a certain audience and also to design the next-generation of sustainable materials with improved properties. This is a topic of growing interest. However, it exists some points that need to be addressed (please, see them below detailed point-by-point) to improve the scientific quality of the submitted manuscript paper before this article will be consider for its publication in Materials.

1) “Given the intensifying global energy crisis (…) sustainable and energy-efficient” (lines 25-26). The authors should provide quantitative data insights about the worldwide energy consumption global burdens. This will significantly aid the potential readers to better understand the significance of the devoted research.

2) “Cellulose emerges as a promising alternative (…) thermal insulation is curtailed due to inadequate flame retardancy and poor mechanical properties (…) superior elasticity, fatigue resistance, and enhanced flame retardancy” (lines 29-34). Here, even if I agree with the information provided in this statement, it should be also mentioned how the low mechanical properties observed in cellulose materials are caused by the moisture uptake at certain environmental relative humidities [1].

[1] Marcuello, C.; Foulon, L.; Chabbert, B. ; Aguié-Béghin, V. ; Molinari, M. Atomic force microscopy reveals how relative humidity impacts the Young’s modulus of lignocellulosic polymers and their adhesion with cellulose nanocrystals at the nanoscale. Int. J. Biol. Macromol. 2020, 147, 1065-1075. https://doi.org/10.1016/j.ijbiomac.2019.10.074.

3) “Nevertheless, its broader applications in thermal insulation is curtailed due to inadequate flame retardancy (….) for producing cellulose aerogels with superior elasticity, fatigue resistance, and enhanced flame retardancy” (lines 30-34). Similar than the previous point, there lacks a relevant citation reference related to the information provided in this statement.

[2] Fan, B.; Chen, S.; Yao, Q.; Sun, Q.; Jin, C. Fabrication of Cellulose Nanofiber/AlOOH Aerogel for Flame Retardant and Thermal Insulation. Materials 2017, 10, 311. https://doi.org/10.3390/ma10030311.

4) “Bagasse was obtained from (…) Zibo Bin Hong Insulation Materials Co., Ltd” (lines 72-73). Did the authors chemically characterize the materials used in this work before to prepare the tested aerogels?

5) Figure 1 caption (lines 94-97). The information related to the yellow and blue arrows highlighted in the panel b should be stated in the respective figure caption.

6) “The microstructural analysis (…) field emission scanning electron microscope” (lines 118-119). What was the settled electron acceleration voltage to gather the SEM images? Did the authors used any contrast agent? Some information should be furnished in these regards.

7) Finally, the information concerning the software tools to process the raw data needs to be also detailed.

8) “Figure 1f presents the pore size (…) of 58.91 µm” (lines 141-144). Please, the authors should add the standard deviation (SD) associated to the mean pore size value. Same comment for the bars plotted in the histograms of the Figure 3b (line 200) and the Fig. 4a (line 237).

9) Figure 2, panel l (SEM images, line 175). The lateral scale bars should be added in these figures.

10) “3.2. Mechanical properties of Mubce aerogels” (lines 196-229). Are the differences observed in the mechanical properties of bagasse cellulose and mullite fibers reinforced bagasse cellulose aerogels statistically different? Same question for the thermal insulation and fire resistance properties depicted in the subsection 3.3 in the lines (230-294).

11) “4. Conclusions” (lines 295-309). This section perfectly remarks the most relevant outcomes found by the authors in this work and the promising future perspectives. It should be desirable to add a brief statement to discuss about the potential future action lines to pursue the topic of this research.

Author Response

(The authors gave the same response as above.)

Reviewer 3 Report

Comments and Suggestions for Authors

The manuscript titled "Mullite Fibers Reinforced Bagasse Cellulose Aerogels with Excellent Mechanical, Flame Retardant, and Thermal Insulation Properties" is original and has interesting and applicable values. The topic is suitable for the journal, and the research design and methodology are adequate. The English language is at an acceptable level. However, there are some minor issues that decrease the overall quality of the manuscript:

  1. Figures 2a-f contain several small letters that are too small to be readable. I recommend enlarging their size.

  2. The graph in Figure 3b should include error bars for each obtained value.

  3. Figure 4a has the same issue with error bars. Additionally, the Y-axis range would be best from 0.020 to 0.030.

  4. The Discussion and Conclusions sections include some future perspectives; however, they would benefit from a more structured and detailed outline of future work.

Overall, the manuscript is of sufficient interest. The obtained data are interesting and contribute to advancements in the studied subject. However, a few minor issues need to be addressed before the manuscript can be accepted for publication.

Author Response

(The authors gave the same response as above.)
